# Polycyclic Aromatic Hydrocarbons Contamination of Flamed and Braised Chickens and Health Risk Assessment in Burkina Faso

**DOI:** 10.3390/toxics9030065

**Published:** 2021-03-18

**Authors:** Bazoin Sylvain Raoul Bazié, Caroline Douny, Thomas Judicaël Ouilly, Djidjoho Joseph Hounhouigan, Aly Savadogo, Elie Kabré, Marie-Louise Scippo, Imaël Henri Nestor Bassole

**Affiliations:** 1Laboratoire de Biologie Moléculaire d’Epidémiologie et de Surveillance des Agents Transmissibles par les Aliments (LABESTA), Unité de Formation et de la Recherche en Sciences de la Vie et de la Terre, Ecole Doctorale Sciences et Technologies, Université Joseph Ki-Zerbo, 03 BP 7021 Ouagadougou 03, Burkina Faso; raoulbazie@gmail.com (B.S.R.B.); judicare2@gmail.com (T.J.O.); 2Laboratoire National de Santé Publique (LNSP), 09 BP 24 Ouagadougou 09, Burkina Faso; elie.kabre@gmail.com; 3Laboratoire d’Analyse des Denrées Alimentaires, Centre de Recherche FARAH—Secteur Santé Publique Vétérinaire, Université de Liège, Bât B43b, Sart Tilman, Boulevard de Colonster, 20, B-4000 Liège, Belgique; cdouny@uliege.be (C.D.); mlscippo@uliege.be (M.-L.S.); 4Laboratoire de Sciences des Aliments, Faculté des Sciences Agronomiques, Université d’Abomey Calavi, 03 BP 2819 Jéricho, Benin; joseph.hounhouigan@gmail.com; 5Laboratoire de Biochimie et d’Immunologie Appliquée (LABIA), Unité de Formation et de la Recherche en Sciences de la Vie et de la Terre, Université Joseph Ki-Zerbo, 03 BP 7021 Ouagadougou 03, Burkina Faso; alysavadogo@gmail.com

**Keywords:** braised chicken, flamed chicken, polycyclic aromatic hydrocarbons, risk assessment, Burkina Faso

## Abstract

Charcoal- or wood-cooked chicken is a street-vended food in Burkina Faso. In this study, 15 samples of flamed chicken and 13 samples of braised chicken were analyzed for 15 priority polycyclic aromatic hydrocarbons (PAHs) with a high-performance liquid chromatography-fluorescence detector. A face-to-face survey was conducted to assess the consumption profiles of 300 men and 300 women. The health risk was assessed based on the margin of exposure (MOE) principle. BaP (14.95–1.75 μg/kg) and 4PAHs (BaP + Chr + BaA + BbF) (78.46–15.14 μg/kg) were eight and five times more abundant at the median level in flamed chickens than in braised ones, respectively. The contents of BaP and 4PAHs in all flamed chicken samples were above the limits set by the European Commission against 23% for both in braised chickens. Women had the highest maximum daily consumption of both braised (39.65 g/day) and flamed chickens (105.06 g/day). At the estimated maximum level of consumption, women were respectively 3.64 (flamed chicken) and 1.62 (braised chicken) times more exposed to BaP and 4PAHs than men. MOE values ranged between 8140 and 9591 for men and between 2232 and 2629 for women at the maximum level of consumption of flamed chickens, indicating a slight potential carcinogenic risk.

## 1. Introduction

The International Agency for Research on Cancer (IARC) classified processed meat in Group I of human carcinogens in 2015 [1]. This was due to the numerous chemical compounds found in the processed meat that potentially threaten consumers’ health. Polycyclic aromatic hydrocarbons (PAHs) represent the major class of such compounds. PAHs are composed of two or more combined aromatic rings and have a carcinogenic effect [2]. These compounds are generated during the incomplete combustion or pyrolysis of hydrocarbons such as coal, gas, oil and wood [3]. They are nonpolar compounds with lipophilic, semi-volatile and persistent properties [4,5,6]. They were evaluated by the IARC and have been classified into Group I for benzo[a]pyrene and Group 2A, Group 2B and Group 3 for the other members of this family of compounds [4]. Therefore, they are listed as priority pollutants by the European Union and the United States Environmental Protection Agency (US EPA) [7]. Some organizations, such as the International Programme on Chemical Safety, the Scientific Committee on Food and the Joint FAO/WHO Expert Committee on Food Additives, concluded that 15 PAHs, namely benz[a]anthracene (BaA), benzo[b]fluoranthene (BbF), chrysene (CHR), benzo[j]fluoranthene (BjF), benzo[k]fluoranthene (BkF), benzo[ghi]perylene (Bgp), benzo[a]pyrene (BaP), cyclopenta[cd]pyrene (CcP), dibenz[a,h]anthracene (DhA), dibenzo[a,e]pyrene (DeP), dibenzo[a,h]pyrene (DhP), dibenzo[a,i]pyrene (DaP), dibenzo[a,l]pyrene (DlP), indeno[1,2,3-cd]pyrene (IcP) and 5-methylchrysene (5MC) are potentially toxic to humans and should be a priority group in the assessment of the risk of long-term adverse health effects following dietary intake [8]. Various approaches have been used to characterize the risk related to PAH exposure. The toxic equivalency factor (TEF) has been described by several studies [9,10,11,12]. The use of the TEF approach is based on a principle assuming that the individual compounds act through the same biologic or toxic pathway, the effects of individual chemicals in a mixture are essentially additive at submaximal levels of exposure, the dose-response curves for different congeners are parallel and the organotrophic manifestations of all congeners are identical over the relevant range of doses [13]. However, the European Food Safety Authority found that there are several problems with using the TEF approach in the risk assessment of PAHs in food and proposed the margin of exposure (MOE) approach for risk characterization [8].

Humans are exposed to PAHs by inhalation and skin contact as well as through consumption of contaminated food. Consumption of contaminated food has been identified as the principal mode of exposure, contributing to about 88 to 98% [14]. Numerous authors have demonstrated that dietary exposure to PAHs is associated with an increased risk of some human cancers [15,16,17,18].

Processing techniques such as roasting, barbecuing, grilling, smoking, heating, drying and baking contribute towards PAH formation [5]. The concentrations of PAHs in processed meat are dependent on a number of processing parameters including distance to heat source, combustibles, level of processing and cooking time and methods [19].

In Burkina Faso, locally bred chickens, known as poulet bicyclette, are traditionally processed into braised or flamed chickens by direct exposure to burned charcoal and wood flame. Flamed and braised chickens are processed and sold as street foods. They are popular in the population meat consumption behavior. A number of authors have reported contamination of processed street-vended meat in Burkina Faso by pathogens [20,21,22] and metallic trace elements [23]. However, there are few data on PAH contamination of processed chicken in Burkina Faso. Furthermore, the risk assessment of dietary exposure to PAHs in grilled chicken products in Burkina Faso remains unknown. Therefore, the aims of this study were to determine 15 priorities PAHs contents in flamed and braised chickens in Burkina Faso and assess human health risks. 

## 2. Materials and Methods

### 2.1. Reagents and Standards

Individual standards of PAH (BaA, BbF, BjF, BkF, BgP, BaP, Chr, DhA, DeP, DhP, DiP, DlP, IcP, 5MC and BcL) solutions in acetonitrile (with purity varying from 98.5% to 99.9%) were provided by Cluzeau Info Labo (Putteaux la Défense, France). Deuterated DiP-D14 in toluene with 99.7% purity (LGC Promochem, France) was used as internal standard. The HPLC and GC-MS graded solvents (dichloromethane, hexane, acetone, ethyl acetate, cyclohexane, ethanol, acetonitrile) were obtained from Biosolve (Valkenswaard, The Netherlands) and VWR (Leuven, Belgium). Florisil (Promochem, Germany) and celite (Fluka, Sigma-Aldrich, St Louis, MI, USA) were used for the accelerated solvent extraction.

### 2.2. Apparatus

The extraction of the PAHs was done using an Accelerated Solvent Extraction system (Dionex 200, Sunnyvale, CA, USA). A TurboVap^®^ II evaporator (Zymarck, Germany) was used for evaporation. The samples analysis was carried out using a Waters Model 600 E HPLC system (Milford, MA, USA), equipped with photo diode array (PDA) and fluorescence detectors. A C18 Pursuit 3 PAH column (100 mm × 4.6 mm, 3 μm) equipped with a ChromGuard (10 × 3 mm) precolumn (Varian Inc., Palo Alto, CA, USA) was used to separate the PAHs.

### 2.3. Samples Collection

Twenty-eight (28) ready-to-eat samples of processed chickens, including fifteen (15) samples of flamed chicken and thirteen (13) samples of braised chicken, were considered. The samples were collected in the city of Ouagadougou (Burkina Faso) from randomly selected sellers and selling points. The samples were collected in sterile stomacher bags and carried to the laboratory in an icebox. Braised and flamed chickens were processed using burned charcoal and wood flame as combustible respectively. The processing parameters of each type of chicken, including the nature of the fuel, cooking temperature, distance to the heat source, mass loss and cooking time, were recorded.

### 2.4. Samples Pre-Treatment

The chicken were deboned, ground and stored in a freezer (Liebherr, Germany) prior to analyses. The samples were dried using a dryer (Memmert, Büchenbach, Germany), according to the AOAC 950.46 method [24]. Briefly, 5 g (+/−0.001 g) of samples were weighed using aluminum dishes and placed into an oven maintained at 103 °C. The dishes were removed from the oven and placed into the desiccator to cool at room temperature and weighed to the nearest 0.001 g. Heating, cooling and weighing were repeating as long as a constant weight was obtained.

### 2.5. Sample Extraction

The PAHs were extracted using an Accelerated Solvent Extraction (ASE) system as described by Kpoclou and co-workers [25]. The ASE cells were conditioned with 0.5 g of celite (Celite Filter Cel, Fluka, Sigma-Aldrich, St Louis, MI, USA) and 7.5 g of florisil (Promochem, Germany) and then washed with dichloromethane prior to extraction. One gram (1 g) of dried sample was weighed in the cell and extracted with 6 mL of hexane/acetone (50:50, *v*/*v*). The extract was evaporated under nitrogen stream until there was 1 mL remaining. The concentrate extract solution was reconstituted with 5 mL of cyclohexane.

### 2.6. Sample Purification

The reconstituted extract was cleaned up using solid-phase extraction columns (Macherey-Nagel Chromabond). The column was conditioned with 15 mL of ethyl acetate and 10 mL of cyclohexane. The extract was then loaded into the column. After loading, the column was washed with 6 mL of cyclohexane/ethanol (70:30, *v*/*v*). The PAHs were then eluted with 12 mL of cyclohexane/ethyl acetate (40:60, *v*/*v*). The eluate was evaporated to dryness and the residue was dissolved in 90 μL of acetonitrile and then spiked with 10 μL of deuterated DiP and transferred into a vial prior to HPLC analysis.

### 2.7. HPLC/FLD Analysis

The chromatographic conditions were set following Brasseur et al. (2007) [26]. The injection volume and the column temperature were 5 μL and 25 °C, respectively. The separation was performed with the mobile phase gradient as presented in the Table 1.

Seven calibration solutions varying from 10 to 1600 pg/μL (BjF and IcP) and 2.5 to 400 pg/μL (the other PAHs) were used. A quadratic regression was performed for curve fitting and calculation of the native PAH concentration was based on the ratio between native and internal standard PAH peak areas. Three detection channels were set to optimize the detection (Table 2).

### 2.8. Quality Control

A blank procedure and a spiked blank matrix were injected with the samples series to check the process of extraction and purification. They were used to calculate the recovery for each PAH. A certified solution containing the 15 PAHs at 20 pg/μL individually was injected with samples series. The limits of quantidication (LOQs) and the variation of the retention time were determined.

### 2.9. Exposure Assessment

Since there were no available data on the consumption of processed chicken, especially braised and flamed chicken, at the national level, a face-to-face survey was designed with Sphinx (Plus2 Version 5) to collect data on the consumption of processed chicken for the population of Ouagadougou. The questionnaire is provided in the Appendix A (Appendix A). The configuration of the participants of the survey was designed according to the socioprofessional group distribution of the population of Ouagadougou. A sample of 600 adults (17–62 years old), including 300 men and 300 women, were interviewed. The participants were asked to provide the frequency of their processed chicken consumption and their consumption habits. The collected data for both frequencies and the individually consumed ratio (chicken is generally eaten in groups in Burkina Faso) were reported to the average weight of each type of processed chicken in order to obtain the estimated daily consumption of processed chickens expressed as g/day.

A deterministic approach was adopted to evaluate the dietary exposure. At the minimum level of processed chicken consumption, the exposition to genotoxic PAHs was too low; therefore, the maximum and median levels of consumption were considered with the PAHs median concentration in the samples.

The estimated dietary intake of PAHs expressed as estimated daily intake (EDI) (ng/kg body weight/day) was determined using the following equation [8]:(1)$$\mathrm{EDI}=\frac{C\;\times \;\mathrm{IR}}{\mathrm{BW}}$$
where C is the median content of PAHs in the samples (μg/kg), IR is the daily consumption rate of chicken (kg/day) and BW is the body weight (kg) (men = 65.2 kg and women = 59.0 kg in Burkina Faso) [27].

### 2.10. Risk Assessment

The margin of exposure was used to characterize the risk from the consumption of processed chickens. It was assumed that a margin of exposure of 10,000 or higher would mean that the consumer exposure was of low concern from the viewpoint of public health [8]. A benchmark dose lower confidence limit of 10% (BMDL10), the estimated lowest dose, including 95% confidence interval causing no more than a 10% incidence of cancer in rodents, was used to calculate the MOE. The margin of exposure for each consumer group was calculated as follows [8]:(2)$$\mathrm{MOE}=\frac{\mathrm{BMDL}10}{\mathrm{EDI}}$$
where BMDL10 is the benchmark dose lower confidence limit at 10% incidence level and EDI is the estimated daily intake. BMDL10 of 0.07, 0.17, 0.34 and 0.49 in mg/kg bw/day was considered for BaP, 2PAHs (BaP + Chr), 4PAHs (BaP + Chr + BaA + BbF) and 8PAHs (BaP + Chr + BaA + BbF + BkF + BgP + DhA + IcP), respectively.

### 2.11. Statistical Analysis

Statistical analyses were performed using SPSS software, version 23.0.0.0. The data were tested for normal distribution (Shapiro–Wilk test) and homogeneity (Levene test). A Mann–Whitney U test was used to assess the differences between the groups considered significant at *p* < 0.05.

## 3. Results and Discussion

### 3.1. The Quality Assurance

The recovery rates ranged between 59.07% and 118.87%, which is in agreement with the performance criteria of the European Commission Regulation (EC) N° 333/2007 [28]. The relative retention time of all PAHs detected in samples complied with the criteria of variation of ±2.5% [29] compared to the relative retention time of the PAHs standard. The LOQs of the method were 0.1 μg/kg for BaP, Chr, BbF, BaA, Bkf, BgP, DhA, Dep, 5MC, DlP, DiP, BcL and DhP, and 0.5 μg/kg for IcP and BjF.

### 3.2. Concentration of PAHs in the Processed Chicken

Since the contents of PAHs failed to show a normal distribution (Shapiro-Wilk test, *p* < 0.05), the results are presented as the minimum, the maximum and the median concentrations followed by the interquartile range (IQR). The concentrations of the 15 PAHs in both braised and flamed chickens are shown in Table 3. BaP (14.95 and 1.75 μg/kg) and 4PAHs (BaP + Chr + BaA + BbF) (78.46 and 15.14 μg/kg) were respectively eight and five times more abundant at the median level in flamed chickens than in braised ones. The median concentrations of both BaP and 4PAHs were higher than those found in traditional Lebanese grilled chicken [30], Iranian charcoal-grilled chicken [31] and Turkish grilled chicken [32]. The contents of BaP and 4PAHs in all flamed chicken samples were above the limits set by the European Commission [33] against 23% for both in braised chickens. Charcoal and wood were used as combustibles in the processing of braised and flamed chickens, respectively. The distances between the heat source and the braised and flamed chickens were 5 ± 2 and 3 ± 1 cm respectively. The average cooking time was 45 ± 10 min for braised and 25 ± 5 min for flamed chickens. Both braising and flaming heat temperatures were above 500 °C. Previous studies have shown that the highest concentrations of PAHs in barbecued meat were due to cooking over open flames [34], the closeness of the meat to the heat source [35], the type of fuel used [36] and too short a cooking time at high temperature [37]. Additionally, it has been observed that, during the cooking of both braised and flamed chickens, the producers coated the outer surfaces of the chicken with oil, which was shown to increase the formation of PAHs [38]. The highest concentrations of PAHs in flamed chicken could be explained by the combined effect of wood flame, high temperature, addition of oil during the process, the close contact with the energy source and the short cooking time. The variation observed in the PAH concentrations in both flamed and braised chicken could have also been due to certain parameters not recorded in this study, such as the fat content of the raw chicken [35,39] and the type of wood and charcoal used as combustibles [40], which can affect PAH formation.

### 3.3. Human Health Risk Assessment of PAHs in Braised and Flamed Chicken

#### 3.3.1. Estimation of Daily Consumption of Processed Chicken

The median daily consumptions of braised and flamed chicken were respectively 3.97 g/day and 5.25 g/day for both women and men (Table 4). The women group had the highest maximum daily consumption of both braised (39.65 g/day) and flamed (105.06 g/day) chickens. Considering the maximum level, women’s daily consumption of flamed and braised chicken were 3.30 and 1.60 times higher than those of men. The results obtained in the estimation of the daily consumption of processed chickens show a great variability. This variability could be due to the heterogeneous character of the population in its socioprofessional constitution, which is a major determinant in the consumption of luxury foodstuffs such as meat and meat products.

#### 3.3.2. Estimation of Daily Intake of Genotoxic PAHs

The estimated daily intakes of BaP, 2PAHs, 4PAHs and 8PAHs were evaluated for the maximum and median daily consumption of processed chicken (Figure 1). The highest maximum EDIs of BaP (26.62 ng/kg bw/day), 2PAHs (76.18 ng/kg bw/day), 4PAHs (139.71 ng/kg bw/day) and 8PAHs (212.56 ng/kg bw/day) were recorded for women’s flamed chicken consumption. At the estimated maximum level of consumption, women were 3.64 (for flamed chicken) and 1.62 (for braised chicken) times more exposed to genotoxic and carcinogenic polycyclic aromatic hydrocarbons than men. However, at the estimated median consumption for both flamed and braised processed chicken, no significant differences were observed for men and women groups. The estimated daily intakes of BaP, 4PAHs, 8PAHs and 16PAHs in the Turkish population were estimated to be 0, 1.79, 3.09 and 4.12 ng/kg bw/day from grilled chicken consumption [32]. These EDI are lower than those found in the current study. Jiang and colleagues [9] reported exposure of adult Chinese to 0.49, 3.96, 4.99 and 120 ng/kg bw/day respectively for BaP, 4PAHs, 8PAHs and 15PAHs from grilled meat consumption. It is assumed that the dietary intake of PAHs is in relation with the dietary habits of the consumers and the contamination levels of PAHs in foods. Also it has been stated that the consumers would be exposed to more than average levels of PAHs as the consumption rate increases [32].

#### 3.3.3. Risk Characterization

The margin of exposure is the ratio between a benchmark dose (BMDL_10_) and the estimated daily intake of a given population. There is a concern for consumer health when the margin of exposure is lower than 10,000 [8]. The margins of exposure for BaP, 2PAHs, 4PAHs, and 8PAHs were evaluated for each consumer group and are presented in the Figure 2. At the median and maximum consumption levels of braised chicken and the median consumption level of flamed chicken, the MOE to BaP, 2PAHs, 4PAHs, and 8PAHs were higher than 10,000 and varying between 656,927 and 33,589, indicating a very low exposure to cancer risks. However, at the maximum level consumption of flamed chicken, the results indicate that the MOE for BaP, 2PAHs, 4PAHs and 8PAHs were lower compared to the critical value of 10,000 (Figure 2) set by the European Food Safety Authority (EFSA), which means a higher threat to both women and men consumer’s health.

The low MOE values obtained for flamed chicken at the maximum level of consumption might be explained by the high concentrations of PAHs and daily consumption (Figure 1). Since this is associated with possible exposure to cancer risk [1], it is worth paying attention to daily flamed chicken consumption.

## 4. Conclusions

The results of the present study showed that braised and flamed chicken were highly contaminated by PAHs. The median concentrations of PAHs in flamed chickens were higher than those of braised chickens. Women were 3.64 to 1.62 times more exposed to PAHs than men. The cancer risk assessment showed MOE values below the safety threshold of 10,000 for flamed chicken, at the estimated maximum consumption level for both men and women consumer groups, indicating a public health concern. Braised and flamed chicken processing techniques must be improved to reduce such contamination and the relevant measures must be taken by competent authorities to protect consumers.

## Figures and Tables

**Figure 1 toxics-09-00065-f001:**
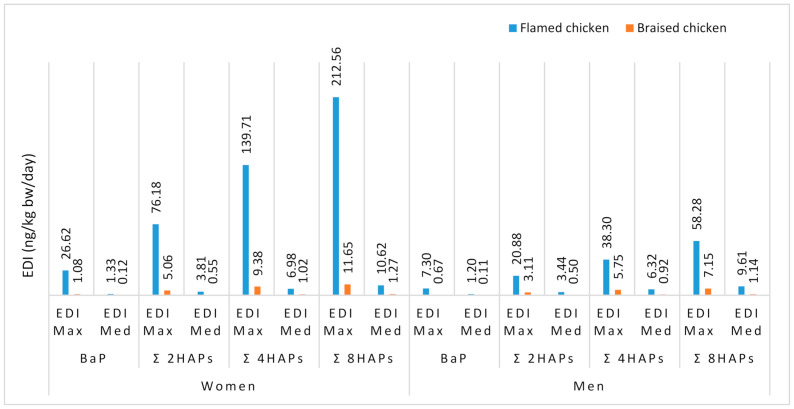
Daily intake of genotoxic PAHs (ng/kg bw/day).

**Figure 2 toxics-09-00065-f002:**
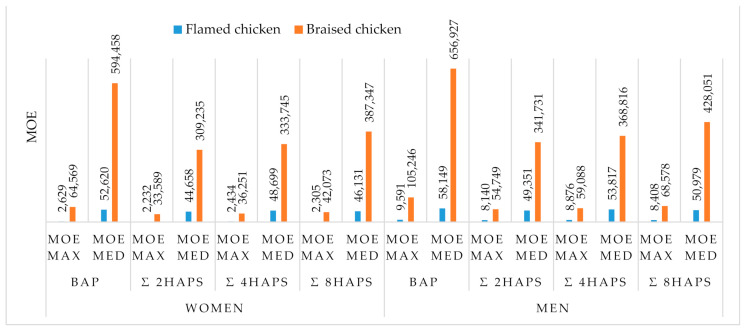
Margins of exposure for BaP, 2PAHs, 4PAHs and 8PAHs.

**Table 1 toxics-09-00065-t001:** Mobile-phase elution gradient.

Time (min)	Flow (mL/min)	Water (%)	Acetonitrile (%)	Methanol (%)
0	1	15	30	55
2	1	15	30	55
20	1	0	100	0
25	1.5	0	100	0
40	1.5	0	100	0
45	1	15	30	55

**Table 2 toxics-09-00065-t002:** Wavelengths detection of 15 European Union priority polycyclic aromatic hydrocarbons (PAHs).

	Excitation (nm)	Emission (nm)	PAHs
Channel A	280	438	BbF
	285	420	DlP, DhA, Bgp, DeP
Channel B	300	512	BjF
Channel C	304	360	Bcl
	275	385	BaA, CHR, 5MC,
	280	405	BkF, BaP
	280	499	IcP
	380	434	Dip-D14, DiP
	290	450	DhP

**Table 3 toxics-09-00065-t003:** Concentrations of PAHs in processed chicken (μg/kg).

PAH	Braised Chicken (*n* = 13)			Flamed Chicken (*n* = 15)		
	Min	Max	Med (IQR)	Min	Max	Med (IQR)
BaP	0.5	4.62	1.75 (1.93) ^b^	3.86	40.19	14.95 (25.25) ^a^
Chr	1.89	29.56	6.69 (5.61) ^b^	16.65	81.17	33.03 (37.54) ^a^
BaA	2.33	12.41	4.65 (4.23) ^b^	9.49	68.21	19.33 (21.71) ^a^
BbF	0.78	6.04	2.40 (2.71) ^b^	3.96	56.18	12.66 (20.47) ^a^
BkF	0.18	1.7	0.65 (0.63) ^b^	1.12	32.23	5.60 (11.27) ^a^
BgP	0.78	3.5	1.64 (1.55) ^b^	2.5	61.61	14.10 (19.00) ^a^
DhA	0.04	0.8	0.25 (0.17) ^b^	0.41	14.6	3.39 (4.28) ^a^
IcP	0.19	8.39	0.80 (0.59) ^b^	1.24	43.4	6.13 (11.89) ^a^
DeP	ND	0.77	0.31 (0.52) ^b^	0.49	8.27	2.18 (3.69) ^a^
BjF	0.34	4.56	1.56 (1.91) ^b^	ND	62.71	8.33 (11.26^) a^
BcL	ND	10.71	4.38 (4.31) ^a^	5.11	23.26	11.34 (10.03) ^a^
DlP	ND	0.1	0.00 (0.00) ^a^	ND	3.1	0.00 (0.00) ^a^
5MC	ND	13.44	0.00 (0.27) ^a^	ND	1.52	0.00 (0.60) ^a^
DiP	ND	0.19	0.00 (0.00) ^b^	ND	2.74	0.35 (0.89) ^a^
DhP	ND	1.24	0.00 (0.00) ^b^	ND	1.01	0.17 (0.40) ^a^
2PAHs	3.11	34.18	8.17 (6.35) ^b^	20.51	118.29	42.78 (50.39^) a^
4PAHs	6.98	46.94	15.14 (12.88) ^b^	33.96	226.10	78.46 (93.59) ^a^
8PAHs	8.19	61.28	18.80 (16.31) ^b^	39.24	331.74	119.37 (171.63) ^a^
15PAHs	10.26	78.43	24.48 (17.66) ^b^	48.63	429.4	146.94 (193.53) ^a^

2PAHs: benzo[a]pyrene + (BaP)+ Chrysene (Chr); 4PAHs: benzo[a]pyrene (BaP)+ Chrysene (Chr)+ benz[a]anthracene (BaA)+ benzo[b]fluoranthene (BbF); 8PAHs: benzo[a]pyrene (BaP)+ Chrysene (Chr)+benz[a]anthracene (BaA)+ benzo[b]fluoranthene (BbF)+ benzo[k]fluoranthene (BkF)+ benzo[ghi]perylene (BgP)+dibenz[a,h] anthracene (DhA)+ indeno[1,2,3-cd]pyrene (IcP); Min: minimum, Max: maximum, Med: median, IQR: interquartile range, ND: not detected. ^a,b:^ Medians in the same row followed by the same superscript letters indicate no significant difference (*p* < 0.05) according to Mann–Whitney U tests.

**Table 4 toxics-09-00065-t004:** Estimated daily consumption of braised and flamed chicken (g/day) [23].

Type	Parameters	Men (*n* = 300)	Women (*n* = 300)
Braised chicken	Minimum	0.62	0.62
	Maximum	24.78	39.65
	Median (IQR)	3.97 ^a^ (4.13)	3.97 ^a^ (7.93)
Flamed chicken	Minimum	1.05	1.05
	Maximum	31.83	105.06
	Median (IQR)	5.25 ^a^ (7.88)	5.25 ^a^ (7.88)

^a^ Medians in the same row followed by the same superscript letters indicate no significant difference (*p* < 0.05) according to Mann-Whitney U tests.

## Data Availability

The data presented in this study are available on request from the corresponding author.

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
