# Peer review of "Polycyclic Aromatic Hydrocarbons Contamination of Flamed and Braised Chickens and Health Risk Assessment in Burkina Faso"

_toxics, 2021, doi:10.3390/toxics9030065_

Round 1

Reviewer 1 Report

Great job! Nice study! A few minor suggested changes below. 

Pre-treatment of samples (Sentence/Line 89)

"The muscles were separated from the bones" could be rephrased as "The meat was separated from the bones"

Reagents and standards (Line 101)

"Benzo[c]fluorène (BcL) solutions in acetonitrile (with purity varying from 98,5 to 99,9%)". "98,5 to 99,9%"  should be changed to "98.5 to 99.9%".  

Exposure Assessment (Equation 1, Line 175)

In the numerator, "CxIR" should be "C × IR". Multiplication sign,× , should be used and not the italicized letter, x. OR "C × IR" can simply be written as "C(IR)".

Concentration of PAHs in the processed chicken ((Sentence/Line 218 )

"Though, the highest content of PAHs in flamed chicken could be explained by the combined effect of wood flame, high temperature, addition of oil during the process, the close contact with the energy source and the short cooking time"  is a hanging or incomplete sentence. It is better to remove "though' and make the sentence read as  "The highest content of PAHs in flamed chicken could be explained by the combined effect of wood flame, high temperature, addition of oil during the process, the close contact with the energy source and the short cooking time".

Table 6. Margin of Exposure to BaP, 2PAHs, 4PAHs, 8PAHs (Line 252)

Each MOE Med value (for braised chicken for both men and women) should be written together and not separated. For example, "48839  0" should be written together as "488390", just as they are written for the flamed chicken. It could also be that my version of the document has shifted some of the numbers, but I just wanted to draw your attention just in case. 

General Comments

Your conclusion is valid that "It is therefore worth decreasing the daily flamed chicken consumption to a minimum that could help reduce relative cancer risk in the consumers' group". However, you and I know that it is not in human nature to intentionally reduce eating a particular delicacy unless we know of some health hazards associated with it. The thought of ill health or potential death is enough to deter people to reduce their consumption. Therefore, I would suggest discussing the findings/data in this study with some key policy makers in the government. As important as your findings are, it would not be enough to just publish this paper and be done with it. I believe you, the authors of this study, can do more to bring to light the dangers of too much consumption of flamed chicken in your country.  

This findings in this study would also be important to the natives of neighboring countries like Ghana and Togo, and in fact all African countries because I believe flamed chicken is done in similar ways. 

Suggest further research/study.

The authors stated in their Results and Discussion that, "The median concentration of both BaP and 4PAHs were higher than those found in traditional Lebanese grilled chicken, Iranian charcoal grilled chicken and Turkish grilled chicken".

Therefore, an important question that is begging for an answer is, why are the PAHs levels in flamed chicken  in Burkina Faso higher than that  in Lebanese, Iranian, and Turkish grilled/flamed chicken? 

I know the authors are speculating that "The highest content of PAHs in flamed chicken could be explained by the combined effect of wood flame, high temperature, addition of oil during the process, the close contact with the energy source and the short cooking time". However, in science, speculation is not enough when you can have real, hard-core data to support or test out a hypothesis, although in this case  there are previous studies in other countries that support the authors' speculation. I think the authors can source local funding (or otherwise) from the government of Burkina Faso or some interested agencies for such a follow up study. I believe taking real and practical steps to address the situation is a more pragmatic way of expanding on the findings in this study.  Without government intervention, it would be very difficult to tell the natives of Burkina Faso to reduce their daily intake of their most-preferred and delicious flamed chicken. 

Great job on the paper!

Author Response

Pre-treatment of samples (Sentence/Line 89)

The muscle has been changed by the meat

Reagents and standards (Line 101)

98,5 to 99,9% have been to 98.5 to 99.9%

Exposure Assessment (Equation 1, Line 175)

CxIR has been changed by C x IR

Concentration of PAHs in the processed chicken ((Sentence/Line 218 )

"Though" has been removed

Table 6. Margin of Exposure to BaP, 2PAHs, 4PAHs, 8PAHs (Line 252)

The table rows and Colums have adjusted to fit all the numbers

General Comments / Suggest further research/study

The results of this study will be brought to the attention of the national food safety system such as the National laboratory Public Health and the Agency of Standardization of Burkina Faso, for the adequate measures to be observed for the proces sing of braised and flamed chickens.

Also as a further research, investigations have been done to improve the processing technics by using of appropriate grilling equipment in order to reduice the PAHs contamination.

Reviewer 2 Report

The paper addresses a relevant topic which is of interest for the risk assessors in PAHs dietary exposure. However, the paper has several and significant shortcomings, as outlined in the below points:

  1. The abstract should include results presented scientifically, e.g. including statistical significance where appropriate, sample size etc.
  2. The introduction should include more background information, e.g. discuss the dietary risk assessment methodologies.
  3. Materials and methods should start with materials (e.g., list of chemicals used) and apparatus and follow by an accurate description of methods should follow—missing information, such as “where”, “when” and “how” the sample activities was carried out were noted.
  4. The rationale behind the methods used should be described, E.g., why the authors chose the standard concentrations ranges reported in line 140-141.
  5. Methods should not include results; please refer to Section 2.8.
  6. Sample size should be reported accurately as well as the interval of confidence and SD where appropriate. Overall, there is too much deviation in the reported results, especially on processed chicken's daily consumption and in the Concentration of PAHs in processed chicken.
  7. Questionnaire mentioned by the Authors in line 166 should be reported (even as supplementary information). The description appears too vague and poses concerns over the accuracy of the questionnaire design.
  8. Reference to conventional dietary exposure and related risk assessment methods is missing and poses concern over the researchers' methodology.
  9. The rationale behind the chosen statistical analysis should be discussed. The statistical analysis results mentioned in the “methods” section should be adequately reported and discussed in the “Result and Discussion” section.
  10. Line 242 – 243: the SD is not reported here. It is quite unlikely that the cocking distance from heat sources is the same in each sample. Also, the type of heat sources should have been reported. Same applies for the cocking time. It would be useful to know any deviation from the time to gain more insight into the results.
  11. Tables 5&6: The tables' layout and results presentation should be improved to make them more accessible. Also, statistical analysis results should be reported in tables 5 and 6. Statistical results should be fully included here and not in the subscription. Authors should consider graphs were appropriate.
  1. The conclusion should be carefully revised. Other than reducing the flamed chicken consumption, other more effective remediations could be suggested here.

As such, the paper needs significant revisions and more work. Once the shortcoming is fully addressed, it can be reconsidered for submission.

Author Response

Reviewer 2

The abstract should include results presented scientifically, e.g. including statistical significance where appropriate, sample size etc

The abstract has been revised. The sample and the concentrations size have been detailed.

The introduction should include more background information, e.g. discuss the dietary risk assessment methodologies

An additional paragrah about dietary risk assessment methodologies has been included in the introduction part.

Materials and methods should start with materials (e.g., list of chemicals used) and apparatus and follow by an accurate description of methods should follow—missing information, such as “where”, “when” and “how” the sample activities was carried out were noted

The order of the subtitles of the part of Materials and methods have been changed, starting by chemicals used which is Reagents and standards, followed by the apparatus used. More details on sampling process have been given, including the size, the area, the collection condition.

The rationale behind the methods used should be described, E.g., why the authors chose the standard concentrations ranges reported in line 140-141

The analytical methodologies have been more decribed with some details explaining the options used.

Methods should not include results; please refer to Section 2.8

The obtained results on the Quality control issue have been placed in the Results and discussion section

Sample size should be reported accurately as well as the interval of confidence and SD where appropriate. Overall, there is too much deviation in the reported results, especially on processed chicken's daily consumption and in the Concentration of PAHs in processed chicken.

The reporting of the samples size have revised. However the large deviation observed is link to the data (concentration of PAHs and and daily consumption) distribution since the data were not homogen (levene test) and not followed normal distribution (Shapio- Wilk test). For these reasons the results have been presented by Maximun, Minimum and Median effected to the interquartil range

Questionnaire mentioned by the Authors in line 166 should be reported (even as supplementary information). The description appears too vague and poses concerns over the accuracy of the questionnaire design

This part has been reformulated.

Reference to conventional dietary exposure and related risk assessment methods is missing and poses concern over the researchers' methodology

The relevant references of risk assessment methodology have been cited to support the adopted methodology

The rationale behind the chosen statistical analysis should be discussed. The statistical analysis results mentioned in the “methods” section should be adequately reported and discussed in the “Result and Discussion” section.

The distribution of both data (concentration and consumption) were checked for the homogeneity and normal distribution by performing respectively Levene and Shapiro-Wilk tests. Since the data failed to be normal and non homogen, an non parametric (Mann-Whitney U) test should be done to evaluate the statistically difference.

Line 242 – 243: the SD is not reported here. It is quite unlikely that the cocking distance from heat sources is the same in each sample. Also, the type of heat sources should have been reported. Same applies for the cocking time. It would be useful to know any deviation from the time to gain more insight into the results

 The processing parameters such as time, distance to the heat source  have been affected by the standard deviation to be more reliable

Tables 5&6: The tables' layout and results presentation should be improved to make them more accessible. Also, statistical analysis results should be reported in tables 5 and 6. Statistical results should be fully included here and not in the subscription. Authors should consider graphs were appropriate

The layout of the tables 5 and 6 have been revised,

The conclusion should be carefully revised. Other than reducing the flamed chicken consumption, other more effective remediations could be suggested here

The conclusion have been revised

Reviewer 3 Report

Manuscript ID toxics-1068664: “Polycyclic Aromatic hydrocarbons Contamination of Flamed and Braised Chickens and Health Risk Assessment in Burkina Faso”

The work deals with the presence of Polycyclic Aromatic Hydrocarbons (PAHs) in traditional street food in Burkina Faso together with the estimated daily intake of these dangerous chemicals by population. The research is interesting and important as it could help preventing serious diseases due to the incorrect heat treatment of some foods. Nevertheless, there are important unclear aspects and some errors in the manuscript.

Lines 102-103. It is reported that the deuterated DiP-D14 was used as internal standard. But to our knowledge labeled internal standards are used in combination with Mass Spectrometry only (Isotope Dilution). HPLC-Fluorescence cannot distinguish native Dip from Dip-D14. Rightly, none of the seven recent similar works about PAHs with the HPLC-Fluorescence technique mention the labeled internal standards (DOI:10.1016/j.meatsci.2020.108088, DOI:10.1016/j.foodcont.2012.04.034, DOI:10.1016/j.foodcont.2017.04.024, DOI:10.1016/j.chemosphere.2017.06.017, DOI:10.4081/ijfs.2014.1681, DOI:10.1016/j.foodchem.2010.11.097, 10.1016/j.meatsci.2010.05.032)

Lines 228-230. In the manuscript we read: <<The women group had the highest maximum daily consumption of both braised (31.83 g/day) and flamed (105 g/day) chickens>>. But in the Table 4 the maximum daily consumption of braised chickens for women is not 31.83 g/day but 39.65 g/day

Lines 230-231. In the manuscript we read: <<Considering the maximum level, women daily consumption of flamed and braised chicken were 2.65 and 3.30 times higher than that of men>>. But in the Table 4 women maximum daily consumption of flamed and braised chicken is 3.30 and 1.60 times higher than that of men

Lines 152-154. In the manuscript we read: <<Reference solutions in acetonitrile containing the 15 PAHs at three levels of concentrations (1, 10 and 20pg/L, except for BjF and IcP (5, 50 and 100pg/L) were injected in the same injection series>>. In our opinion concentrations in the range 1-100 pg/L are impossible to detect for the Fluorescence detector

Table 5. There seems to be some problems with the Estimated Daily Intakes reported in table 5. If we apply the equation (1) on line 175

EDI = median concentration of PAHs × ingestion rate of chicken / body weight

the values in Table 5 disagree with the other data reported in the manuscript. If we consider, for example, the value of 33.37 ng/kgBW/day (BaP, EDI maximum, flamed chicken, women) also cited by authors on line 237 as “33.77”, it is not clear how this value is obtained. In fact from Table 3 it can be seen that the median concentration of BaP in flamed chicken is 14.95 µg/Kg, from Table 4 it can be seen that the maximum consumption of flamed chicken for women is 105.06 g/day = 0.10506 Kg/day and from line 178 it can be seen that the average body weight for women is 59 kg. Therefore:

EDI = 14.95 (µg/Kg) × 0.10506 (Kg/day) / 59 KgBW = 0.02662 µg/day/KgBW = 26.62 ng/day/KgBW

Instead the value of 33.37 ng/day/KgBW reported in the Table 5 is never obtained by using any combination of min. max. and median BaP concentration and any combination of min. max. and median consumption of flamed chicken for women. The same situation is valid for the other values in Table 5. For example the EDI maximum for women, flamed chicken, 4PAHs should be 139.71 ng/day/KgBW and not 180.20 ng/day/KgBW as reported in Table 5.      

Author Response

Lines 102-103. It is reported that the deuterated DiP-D14 was used as internal standard. But to our knowledge labeled internal standards are used in combination with Mass Spectrometry only (Isotope Dilution). HPLC-Fluorescence cannot distinguish native Dip from Dip-D14. Rightly, none of the seven recent similar works about PAHs with the HPLC-Fluorescence technique mention the labeled internal standards (DOI:10.1016/j.meatsci.2020.108088, DOI:10.1016/j.foodcont.2012.04.034, DOI:10.1016/j.foodcont.2017.04.024, DOI:10.1016/j.chemosphere.2017.06.017, DOI:10.4081/ijfs.2014.1681, DOI:10.1016/j.foodchem.2010.11.097, 10.1016/j.meatsci.2010.05.032)

Numbers of authors are reported the used of Dip-D14 as internal stantads for PAHs analysis by HPLC-Fluorescence : doi:10.1007/s00769-007-0295-0., doi:10.1080/19440049.2014.916422, doi:10.1016/j.aca.2008.11.049, doi: 10.1002/fsn3.1190, https://doi.org/10.1080/19440049.2020.1726502

However our chromatographic conditions were able to distinguish native Dip from Dip-D14 as it is presented in the below chromatogramm

Lines 228-230. In the manuscript we read: <<The women group had the highest maximum daily consumption of both braised (31.83 g/day) and flamed (105 g/day) chickens>>. But in the Table 4 the maximum daily consumption of braised chickens for women is not 31.83 g/day but 39.65 g/day

<<The women group had the highest maximum daily consumption of both braised (31.83 g/day) and flamed (105 g/day) chickens>> has been corrected to <<The women group had the highest maximum daily consumption of both braised (39,65 g/day) and flamed (105 g/day) chickens>>.

Lines 230-231. In the manuscript we read: <<Considering the maximum level, women daily consumption of flamed and braised chicken were 2.65 and 3.30 times higher than that of men>>. But in the Table 4 women maximum daily consumption of flamed and braised chicken is 3.30 and 1.60 times higher than that of men

<<Considering the maximum level, women daily consumption of flamed and braised chicken were 2.65 and 3.30 times higher than that of men>> have been corrected to <<Considering the maximum level, women daily consumption of flamed and braised chicken were 3.30 and 1.60 times higher than that of men>>.

Lines 152-154. In the manuscript we read: <<Reference solutions in acetonitrile containing the 15 PAHs at three levels of concentrations (1, 10 and 20pg/L, except for BjF and IcP (5, 50 and 100pg/L) were injected in the same injection series>>. In our opinion concentrations in the range 1-100 pg/L are impossible to detect for the Fluorescence detector

There was a mistake on the unit, pg/L have been corrected to pg/μl

Table 5. There seems to be some problems with the Estimated Daily Intakes reported in table 5. If we apply the equation (1) on line 175

EDI = median concentration of PAHs × ingestion rate of chicken / body weight

the values in Table 5 disagree with the other data reported in the manuscript. If we consider, for example, the value of 33.37 ng/kgBW/day (BaP, EDI maximum, flamed chicken, women) also cited by authors on line 237 as “33.77”, it is not clear how this value is obtained. In fact from Table 3 it can be seen that the median concentration of BaP in flamed chicken is 14.95 µg/Kg, from Table 4 it can be seen that the maximum consumption of flamed chicken for women is 105.06 g/day = 0.10506 Kg/day and from line 178 it can be seen that the average body weight for women is 59 kg. Therefore:

EDI = 14.95 (µg/Kg) × 0.10506 (Kg/day) / 59 KgBW = 0.02662 µg/day/KgBW = 26.62 ng/day/KgBW

Instead the value of 33.37 ng/day/KgBW reported in the Table 5 is never obtained by using any combination of min. max. and median BaP concentration and any combination of min. max. and median consumption of flamed chicken for women. The same situation is valid for the other values in Table 5. For example the EDI maximum for women, flamed chicken, 4PAHs should be 139.71 ng/day/KgBW and not 180.20 ng/day/KgBW as reported in Table 5.    

Tables 5 and 6 have been revised and corrected

Table 5. Daily Intake of genotoxic PAHs (ng/kg bw/day)

BaP

2PAHs

4PAHs

8PAHs

EDI Maximum

EDI Median

EDI Maximum

EDI Median

EDI Maximum

EDI Median

EDI Maximum

EDI Median

Braised chicken

Men

0,67

0,11

3,11

0,50

5,75

0,92

7,15

1,14

Women

1,08

0,12

5,06

0,55

9,38

1,02

11,65

1,27

Flamed chicken

Men

7,30

1,20

20,88

3,44

38,30

6,32

58,28

9,61

Women

26,62

1,33

76,18

3,81

139,71

6,98

212,56

10,62

Table 6. Margin of Exposure to BaP, 2PAHs, 4PAHs, 8PAHs

BaP

2 PAHs

4 PAHs

8 PAHs

Men

Women

Men

Women

Men

Women

Men

Women

MOE Max

MOE Med

MOE Max

MOE Med

MOE Max

MOE Med

MOE Max

MOE Med

MOE Max

MOE Med

MOE Max

MOE Med

MOE Max

MOE Med

MOE Max

MOE Med

Braised chicken

105246

656927

64569

594458

54749

341731

33589

309235

59088

368816

36251

333745

68578

428051

42073

387347

Flamed chicken

9591

58149

2629

52620

8140

49351

2232

44658

8876

53817

2434

48699

8408

50979

2305

46131

Round 2

Reviewer 2 Report

I thank the Authors for improving their manuscript according to most of my remarks. However, criticalities are still present and need to be fixed prior to publication:

1) The questionnaire used in this work has not been included.

2) Data presentation is reported only using tables, making the results not fully accessible. Please change table 5 and 6 into graphs and consider to include tables in the supplementary information if you wish to retain it. From the numbers it is clear that there is a wide variation within the data, e.g., BAP in Flamed chicken (Woman)-> EDI Median = 1.33, Min 26.62, Max 76.18 (???). Are these results reliable? Please, clarifies as it seemed you had problems in your sampling/analysis activities.

3) On the following Author's comments:

"The reporting of the samples size have revised. However the large deviation observed is link to the data (concentration of PAHs and and daily consumption) distribution since the data were not homogen (levene test) and not followed normal distribution (Shapio- Wilk test). For these reasons the results have been presented by Maximun, Minimum and Median effected to the interquartil range". 

"The processing parameters such as time, distance to the heat source have been affected by the standard deviation to be more reliable" (these claims have to be corrected and rephrased).

You should include these claims in the Results section. You should consider and discuss why your data have such a large deviation, were non-homogeneous and did not follow the normal distribution. Should you consider to repeat the sampling activities with a more robust methodology?

As it stands the manuscript can not be accepted for publication.

Author Response

Responses to the reviewer_Round 3

  • The questionnaire used in this work has not been included.

Please find in the attached files the questionnaire (native form in French) and the Questions translated into English.

The methodology of the face to face survey for processed chicken consumption has been more described in the manuscript.

  • Data presentation is reported only using tables, making the results not fully accessible. Please change table 5 and 6 into graphs and consider to include tables in the supplementary information if you wish to retain it. From the numbers it is clear that there is a wide variation within the data, e.g., BAP in Flamed chicken (Woman)-> EDI Median = 1.33, Min 26.62, Max 76.18(???). Are these results reliable? Please, clarifies as it seemed you had problems in your sampling/analysis activities

Table 5 and table 6 have been changed into graphs.

Table 5 becomes Figure 1

Table 6 becomes Figure 2

The deterministic approach was adopted to evaluate the dietary exposure. At the minimum level of processed chicken consumption, the exposition to genotoxic PAHs was too low, therefore, the maximum and median levels of consumption were considered with the PAHs median concentration in the samples. You may miss read in the table 5 or 6.e.g: BAP in Flamed chicken (Woman)-> EDI Median = 1.33, Max 26.62. There is no Min EDI values reported as you mentioned in your comments

3) On the following Author's comments:

"The reporting of the samples size have revised. However the large deviation observed is link to the data (concentration of PAHs and and daily consumption) distribution since the data were not homogen (levene test) and not followed normal distribution (Shapio- Wilk test). For these reasons the results have been presented by Maximun, Minimum and Median effected to the interquartil range". 

"The processing parameters such as time, distance to the heat source have been affected by the standard deviation to be more reliable" (these claims have to be corrected and rephrased).

You should include these claims in the Results section. You should consider and discuss why your data have such a large deviation, were non-homogeneous and did not follow the normal distribution. Should you consider to repeat the sampling activities with a more robust methodology?

The principle of sampling in this study was based on the diversity of chicken processing habits with variability in the origins of raw chickens, fuel type (wood and charcoal), cooking practices throughout the city of Ouagadougou. The samples were collected from different transformers with a certain variability of these parameters. Since these parameters are not homogeneous it is normal that there is a variation in PAH content in the different analyzed samples because PAHs formation can be linked to these transformation factors.  This may reflect the reality of the contamination levels for each type of chicken processed.

Concerning the consumption data through the survey, the sampling of the participants was based on the socio-professional configuration of the population of Ouagadougou, taking account all social strata. As this sample is not homogeneous on the basis of income, which determines the capacity to consume food, it seems evident that there is some variation in the quantity of chickens consumed.

However, the Results section has been revised to discuss the observed deviation in both PAHs concentration and processed chicken consumption data

Reviewer 3 Report

Abstract, lines 26-28. << The median contents of BaP and 4PAHs in flamed chicken samples were above the limits set by EFSA against 23% for both in braised chickens>>.

The sentence needs to be rephrased.

In fact, in Table 3 we observe that the minimum value for Benzo(a)pyrene was 3.86 µg/kg that is already above the limit. If the minimum value is already above the limit all values for Benzo(a)pyrene were above the limit. The same is valid for 4PAHs, the minimum value of which was 33.96 µg/kg. The limit was established by the European Commission not by EFSA. It is of 2 µg/kg for Benzo(a)pyrene and 12 µg/kg for 4PAHs (Commission Regulation No 835/2011, Annex, Section 6, 6.1.4).

Therefore, the sentence could be rewritten like this:

<< The contents of BaP and 4PAHs in all flamed chicken samples were above the limits set by the European Commission against 23% for both in braised chickens>>.

Abstract, line 30. 1.62 instead of 1.61 in fact on line 238 we read 1.62.

Abstract, lines 31-32. << MOE values ranged from 9591 - 2232 at the maximum level of consumption of flamed chickens for both men and women, indicating a slight potential carcinogenic risk>>.

By examining the Table 6 the sentence should be rewritten as follows:

<< MOE values ranged between 8140 and 9591 for men and between 2232 and 2629 for women at the maximum level of consumption of flamed chickens indicating a slight potential carcinogenic risk>>.

Line 200. 15.14 instead of 15.41 in fact in Table 3 it is reported 15.14

Lines 205-206. Move the reference [33] after <<…the European Commission (EC)…>> in this way: <<…the European Commission (EC) [33]…>>

Line 239. Add <<than men>> after <<…hydrocarbons.>>

Conclusions. Replace <<PAH>> with <<PAHs>> and <<technics>> with <<techniques>>

Reference 33 needs to be replaced, in fact the EC Reg. No 1881/2006 refers only to BaP and it was later amended by the EC Reg. No 835/2011. The right reference is: <<European Commission Regulation No 835/2011 of 19 august 2011 amending Regulation (EC) No 1881/2006 as regards maximum levels for polycyclic aromatic hydrocarbons in foodstuffs. Off. J. Eur. Union L 215/4>>       

Author Response

Responses to the reviewer comments

Abstract, lines 26-28. << The median contents of BaP and 4PAHs in flamed chicken samples were above the limits set by EFSA against 23% for both in braised chickens>>.

The sentence has been rewritten to :

<< The contents of BaP and 4PAHs in all flamed chicken samples were above the limits set by the European Commission against 23% for both in braised chickens>>

Abstract, line 30. 1.62 instead of 1.61 in fact on line 238 we read 1.62.

1.61 has been revised to 1.62

Abstract, lines 31-32. << MOE values ranged from 9591 - 2232 at the maximum level of consumption of flamed chickens for both men and women, indicating a slight potential carcinogenic risk>>

The sentence has been rewritten to :

<< MOE values ranged between 8140 and 9591 for men and between 2232 and 2629 for women at the maximum level of consumption of flamed chickens indicating a slight potential carcinogenic risk>>

Line 200. 15.14 instead of 15.41 in fact in Table 3 it is reported 15.14

15.41 has been revised to 15.14

Lines 205-206. Move the reference [33] after <<…the European Commission (EC)…>> in this way: <<…the European Commission (EC) [33]…>>

The reference [33] has been moved

Line 239. Add <<than men>> after <<…hydrocarbons.>>

 The sentence <<At the estimated maximum level of consumption, women were 3.64 (for flamed chicken) and 1.62 (for braised chicken) times more exposed to genotoxic and carcinogenic polycyclic aromatic hydrocarbons >> has been rewritten to <<At the estimated maximum level of consumption, women were 3.64 (for flamed chicken) and 1.62 (for braised chicken) times more exposed to genotoxic and carcinogenic polycyclic aromatic hydrocarbons than men>>

Conclusions. Replace <<PAH>> with <<PAHs>> and <<technics>> with <<techniques>>

PAH and technics have been replaced by PAHs and techniques

Reference 33 has been replaced by <<European Commission Regulation No 835/2011 of 19 august 2011 amending Regulation (EC) No 1881/2006 as regards maximum levels for polycyclic aromatic hydrocarbons in foodstuffs. Off. J. Eur. Union L 215/4>>

Round 3

Reviewer 2 Report

  1. Thank you for attaching the questionnaire. I hold appropriate to include the questionnaire as Supplementary information. Please refer it in the main text.
  2. Thank you, the graphs are much better than the tables—my apologies for misspelling Min with Max. You might wish to improve the appearance.
  3. The variability of your data remains, but are now properly described and addressed in the manuscript so that the reader can understand the limitations of your study as well as appreciate your results.
  4. Minor English spelling and punctuation missing here and there.

I will accept your manuscript after these minor revisions.

Author Response

Responses to the reviewer comments

  1. Thank you for attaching the questionnaire. I hold appropriate to include the questionnaire as Supplementary information. Please refer it in the main text.

Questionnaire as supplementary information has benn refered in the Materials and method section of the manuscrit

  1. Thank you, the graphs are much better than the tables—my apologies for misspelling Min with Max. You might wish to improve the appearance.

It appear more clear with this graphic mode, thank you for the suggestion

  1. The variability of your data remains, but are now properly described and addressed in the manuscript so that the reader can understand the limitations of your study as well as appreciate your results.

Thank you for your comments on this point, which allowed us to describe with more details the methodology of the sampling and discuss the results by taking account the observed variabilities in both PAHs contents in the samples and processed chicken consumption data. These comments realy improved the quality of the article

  1. Minor English spelling and punctuation missing here and there.

The entire manuscrit has been revised for English spelling and punctuation issues
